

# On the low-energy description for tunnel-coupled one-dimensional Bose gases

**Yuri D. van Nieuwkerk[1,*] and Fabian H. L. Essler[1]**

**1** The Rudolf Peierls Centre for Theoretical Physics, Oxford University, Oxford OX1 3PU, UK

* yuri.vannieuwkerk@physics.ox.ac.uk

## Abstract

We consider a model of two tunnel-coupled one-dimensional Bose gases with hard-wall boundary conditions. Bosonizing the model and retaining only the most relevant inter-actions leads to a decoupled theory consisting of a quantum sine-Gordon model and a free boson, describing respectively the antisymmetric and symmetric combinations of the phase fields. We go beyond this description by retaining the perturbation with the next smallest scaling dimension. This perturbation carries conformal spin and couples the two sectors. We carry out a detailed investigation of the effects of this coupling on the non-equilibrium dynamics of the model. We focus in particular on the role played by spatial inhomogeneities in the initial state in a quantum quench setup.

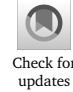
# 1  Introduction

The study of one-dimensional quantum many-body systems out of equilibrium has seen great progress in the past decades. Long-standing questions concerning the equilibration of observables, spreading of correlations and entanglement, and the emergence of statistical mechanics from microscopics have been successfully tackled using a range of innovative theoretical ideas [1–9], whilst spectacular advances in the ability to realize archetypical one-dimensional quantum many-body sytems using cold atoms [10–14] have made it possible to test many of these theoretical developments using tabletop experiments [15–20]. However, such experimental engineering of quantum many-body Hamiltonians relies on certain assumptions to make the experiments map onto a model of physical interest. These assumptions often include having a low energy density, at which an effective low-energy theory holds, and translational invariance, which can generally simplify the problem and specifically play an important role in the integrability of the low-energy theory. When studying non-equilibrium problems in finite quantum many-body systems, these two assumptions are sometimes brought into question.

We here study a situation where both the successes and challenges described above are clearly present: we consider pairs of tunnel-coupled, elongated Bose gases, as realized in the Vienna experiments [13,14,17–19,21–24]. An interesting feature of these experiments is that in certain limits, density measurements after matter-wave interference [13,25] correspond to projective von Neumann measurements of the relative phase field [26]. This allows for the reconstruction of full distribution functions of quantum mechanical observables [21–23], which is of considerable theoretical interest [27–44] in general. In the case at hand, situations without tunnel-coupling can be modelled by a two-component Luttinger liquid [45–47]. This description in terms of a quadratic quantum critical model has yielded theoretical results for the full fluctuation statistics of the relative phase field [28,48,49] which show a satisfying match with experimental results [17,19].

Our interest lies in the effect of a finite tunnel barrier between the gases [14,50–52]. This introduces a relevant perturbation and at sufficiently low energies leads to a decoupled theory of a Luttinger liquid describing the symmetric combination of Bose gas phases ("symmetric sector") and a sine-Gordon model [53] describing the relative phase ("antisymmetric sector"). The sine-Gordon model is of great theoretical importance as it is an exactly solvable, Lorentz invariant quantum field theory that exhibits a rich range of physical phenomena like dynamical mass generation and topological excitations and moreover has important applications to electronic degrees of freedom in solids [54]. Its behaviour out of equilibrium has received a lot of attention in the past decade. To be able to study dynamics, the very weakly interacting limit is

amenable to a simple harmonic approximation [55–57], while the free fermion point can also be used to obtain exact results [55]. Integrability-based methods were used in Refs. [58–61] to study quenches from "integrable" initial states, whereas semiclassical methods [62, 63] were applied to the study of the time-dependence of one and two-point functions as well as the probability distribution of the phase. The truncated conformal space approach [64] was employed in Ref. [65] to analyse the time evolution of two and four-point functions after a quantum quench. A first litmus test for the experimental realization of the sine-Gordon model using split Bose gas experiments was performed in an equilibrium situation: high order equilibrium correlation functions extracted from projective phase measurements in the classical limit have been found to agree well with classical field simulations [23]. For non-equilibrium initial conditions, however, experimental studies [24, 66, 67] have shown puzzling behaviour: when preparing two elongated Bose gases with an initial phase difference, applying a tunnel-coupling between them sets Josephson oscillations of density and phase in motion. These oscillations show a rapid damping, accompanied by a narrowing of the distribution function of the phase. To date, no satisfying theoretical explanation of this damping is known [68]. The damping seems incompatible with a description in terms of a translationally invariant sine-Gordon model, which fails to provide a mechanism for the observed strong and rapid damping in both a self-consistent harmonic treatment [69] and in a combination of truncated Wigner and truncated conformal space approaches [70].

In this work, we go beyond previous studies of the low-energy physics in two important ways:

1. We take into account the next most relevant perturbation at low energies. This perturbation induces an interaction between the symmetric and antisymmetric sectors.

2. We drop the assumption of translational invariance. To this end we place the model in a hard-wall box geometry and consider inhomogeneous initial conditions.

We stress that our focus is on the vicinity of the scaling regime, which is approximately described by a sine-Gordon model as discussed above. This means that we consider energy densities that are small compared to the cutoff of the field theory, which can be taken as the inverse coherence length of the underlying Bose gas. In addition the energy density is of the same order as the mass scale induced by the tunnel-coupling between the Bose gases. Translating these requirements into parameters relevant to the existing experiments gives an energy scale of roughly 5 nK. This is lower than typical energy scales realized in current experimental set-ups, which operate above 18 nK, but sufficiently close to make this regime interesting in light of possible future experiments.

Our strategy is to treat the resulting perturbed sine-Gordon model in the self-consistent time-dependent harmonic approximation (SCTDHA) as described in [69]. In that paper we have benchmarked the approximation using the dynamics of the zero modes in the antisymmetric sector only. It is the expectation value of these modes that displays the Josephson oscillations, making their dynamics vital to the problem at hand. Comparison to numerically exact results for this simplified problem indicated that the SCTDHA offers reliable results for early times corresponding to ∼ 3 density-phase oscillation periods. Based on those findings, we apply the approximation in the current work to the first few oscillation periods in the presence of sector coupling.

We consider the dynamics after initializing the system in a state in which the sectors are uncorrelated and observe how the new coupling term causes correlations between the two sectors to develop over time. In addition to this, energy starts to oscillate between the sectors. Depending on the initial density profile imprinted on the gas, Josephson oscillations of density and phase are affected by the presence of the additional term, showing modulations

of the amplitude that differ from the ones observed in the SCTDHA treatment of isolated sine-Gordon dynamics [69]. However, the observed effects are rather weak. This means that the presence of the box potential and the new sector-coupling term are insufficient to explain the experimentally observed damping phenomenon. At the same time, our results indicate that the simulation of a sine-Gordon model using the hard-wall box potential setup described here should not be severely restricted by the presence of the additional coupling term, which has only mild effects.

This paper is organized as follows. In Sec. 2, we introduce the low-energy effective theory in a box geometry, the additional interaction term and the observable relevant for experiment. We also establish some notational conventions. In Sec. 3, we recapitulate the self-consistent time-dependent harmonic approximation as well as the framework to compute observables and some important distribution functions. In Sec. 4, we apply our formalism to an initial state which is commonly used in the literature, and present results on energy flow and growth of correlations between the sectors, along with the effect on Josephson oscillations, due to the additional interaction term. Sec. 5 summarizes our conclusions and discusses questions for further study.

## 2 Tunnel-coupled Bose gases in a hard-wall box

An appropriate model for the experiments carried out by the Vienna group is an interacting Bose gas confined in three-dimensional space by a tight harmonic potential in the $z$-direction, a double-well potential $V_\perp(y)$ in the $y$-direction and a shallow harmonic potential in the $x$-direction. We will refer to the $x$-direction as *longitudinal*, and to the remaining directions as *transverse*. To simplify the problem, we take the longitudinal potential to be an infinite square well

$$V_\parallel(x) = \begin{cases} 0 & \text{if } 0 < x < L, \\ \infty & \text{otherwise.} \end{cases} \tag{1}$$

Just like a shallow harmonic potential this breaks translational invariance in the longitudinal direction, but it has the additional advantage to be considerably simpler to analyze. Our starting point is thus the following Hamiltonian

$$H_{3d} = \int dx\, dy\, dz\, \left\{ \Psi^\dagger(x,y,z) \left[ -\frac{\nabla^2}{2m} + V_\parallel(x) + V_\perp(y) + \frac{m\omega_z^2}{2}z^2 \right] \Psi(x,y,z) \right.$$
$$\left. + c\left(\Psi^\dagger(x,y,z)\right)^2 \left(\Psi(x,y,z)\right)^2 \right\}, \tag{2}$$

where $\Psi(x,y,z)$ are complex Bose fields obeying the usual bosonic commutation relations.

### 2.1 Low-energy effective theory

In situations where the transverse potentials are sufficiently tight, the dynamics in the $y$- and $z$-directions can be integrated out, in a way analogous to Ref. [71]. Details of this procedure will be reported elsewhere [72]. Projecting to the lowest two states of the transverse potential, and taking appropriate linear combinations of these, we obtain a Hamiltonian for two species

of bosons, $\Psi_{1,2}$, which are approximately localized in wells 1 and 2:

$$H_{\text{1d}} = \int_0^L dx \left[ \sum_{j=1,2} \frac{1}{2m} \partial_x \Psi_j^\dagger(x) \partial_x \Psi_j(x) + \sum_{j,k,l,m=1,2} \Gamma_{jklm} \Psi_j^\dagger(x) \Psi_k^\dagger(x) \Psi_l(x) \Psi_m(x) \right.$$
$$\left. - \left( T_\perp \Psi_1^\dagger(x) \Psi_2(x) + \text{h.c.} \right) \right]. \tag{3}$$

Here the Bose fields $\Psi_i(x)$ have commutation relations $\left[ \Psi_i(x), \Psi_j^\dagger(x') \right] = \delta_{i,j} \delta(x - x')$. The two Bose gases are coupled by a tunnelling term as well as contact interactions. The corresponding coupling constants $\Gamma_{jklm}$ follow from the details of the low-energy projection [72]. For our purposes, we will assume the diagonal elements to be equal to the usual Lieb-Liniger interaction constant, $\Gamma_{jjjj} = g \,\forall\, j$. Hard-wall boundary conditions are imposed by restricting our problem to states $|\Phi\rangle$ where the density at the boundary has a vanishing eigenvalue:

$$\Psi_j^\dagger(L) \Psi_j(L) |\Phi\rangle = \Psi_j^\dagger(0) \Psi_j(0) |\Phi\rangle = 0. \tag{4}$$

The one-dimensional model (3) gives an accurate description of the full theory $H_{\text{3d}}$ at energies that are small compared to the energy $E_{\perp,2}$ of the second excited state of the transverse confining potential. In the actual experiments this is a large energy scale. The physics of interest occurs at energies that are small compared to $v/\xi \ll E_{\perp,2}$, where $\xi$ is the coherence length and $v$ the speed of sound. This enables us to make a second low-energy projection by employing bosonization [45]

$$\Psi_j^\dagger(x) \sim \sqrt{\rho_0 + \partial_x \theta_j / \pi} \, e^{-i\phi_j(x)} \sum_{n=-\infty}^\infty B_n e^{2ni(x\pi\rho_0 + \theta_j)}. \tag{5}$$

This provides a low-energy description of (3) in terms of phase fields $\phi_j$ and $\theta_j$ with a cutoff length scale set by the coherence length of the gases, which for weak interactions is given by $\xi = \pi/mv$ (the sound velocity $v$ is defined below). The hard-wall condition is encoded in the boundary conditions of the $\theta$-fields in a way that is described in Sec. 2.3. Let us first consider the case where interactions and tunnelling between the two gases are absent, meaning that both $T_\perp$ and the non-diagonal elements of $\Gamma$ are zero. This leaves us with two Lieb-Liniger models in a hard-wall box, with interaction strength $g$. Under the mapping (5), the low-energy physics of this model maps to a pair of Luttinger liquids

$$H_j = \frac{v}{2\pi} \int_0^L dx \left[ \frac{1}{K} \left( \partial_x \theta_j(x) \right)^2 + K \left( \partial_x \phi_j(x) \right)^2 \right], \quad j = s, a. \tag{6}$$

Here we have defined (anti)symmetric combinations of the phase fields by

$$\phi_{s/a} = \phi_1 \pm \phi_2, \quad \partial_x \theta_{s/a} = \frac{\partial_x \theta_1 \pm \partial_x \theta_2}{2}. \tag{7}$$

These fields are compact $\phi = \phi + 2\pi$, $\theta = \theta + \pi$ and fulfill commutation relations

$$\left[ \partial_x \theta_j(x), \phi_l(y) \right] = i\pi \delta_{j,l} \delta(x - y). \tag{8}$$

This implies that the canonically conjugate fields to $\phi_j$ are given by

$$\Pi_j(x) \equiv \frac{\partial_x \theta_j(x)}{\pi}. \tag{9}$$

For weak interactions, the sound velocity $v$ and Luttinger parameter $K$ are related to the parameters in the Lieb-Liniger model in a simple way [73]

$$v = \frac{\rho_0}{m}\sqrt{\gamma}\left(1 - \frac{\sqrt{\gamma}}{2\pi}\right)^{1/2}, \quad K = \frac{\pi}{2\sqrt{\gamma}}\left(1 - \frac{\sqrt{\gamma}}{2\pi}\right)^{-1/2}. \tag{10}$$

Here $\gamma = mg/\rho_0$ is the dimensionless interaction strength and $\rho_0$ the average density of each of the two Bose gases.

In the next step we take into account the tunnelling term in (3) as well as "off-diagonal" interaction terms proportional to $\Gamma_{ijkl}$ with not all indices being equal. These introduce relevant perturbations (in the renormalization group sense) with respect to the critical Hamiltonian (6). Inserting the bosonization identity (5) and assuming $\Gamma$ to be real, permutation symmetric and symmetric under $1 \leftrightarrow 2$, we find that the perturbations with the lowest scaling dimensions can be written in the form

$$H_\perp = -2t_\perp \int_0^L dx \, [\rho_0 + \sigma \Pi_s(x)] \cos\phi_a(x), \tag{11}$$

where $t_\perp$ and $\sigma$ depend on the microscopic parameters in (3). Importantly, the two terms in (11) get generated independently and we will therefore treat $t_\perp$ and $\sigma$ as independent phenomenological parameters in the following. The Hamiltonian $H_s + H_a + H_\perp$ should be viewed as the result of integrating out high energy degrees of freedom in a renormalization group sense. As $t_\perp$ grows much faster than $t_\perp \sigma$ under the renormalization group it would be unphysical to consider very large values of $\sigma$. We have therefore restricted the numerical analyses reported below to the range $0 \le \sigma \le 2$. In addition to (11) there are other perturbations with higher scaling dimensions. Their systematic derivation as well as an analysis of their effects will be presented elsewhere [72]. In the case $\sigma = 0$ the full low-energy theory decouples into symmetric and antisymmetric sectors $H = H_s + H_a'$, where $H_a'$ is the Hamiltonian of a quantum sine-Gordon model [53]

$$H_a' = \frac{v}{2\pi}\int_0^L dx \left[\frac{1}{K}(\partial_x \theta_a(x))^2 + K(\partial_x \phi_a(x))^2\right] - 2t_\perp \rho_0 \int_0^L dx \, \cos\phi_a(x). \tag{12}$$

The non-equilibrium dynamics of this model was analyzed for the translationally invariant case in the framework of a SCTDHA in our recent work [69]. The additional $\sigma$-term in (11) couples the sine-Gordon model to the Luttinger liquid Hamiltonian $H_s$. In the following we extend the analysis [69] to

$$H = H_a + H_s + H_\perp. \tag{13}$$

## 2.2 Time-of-flight measurements

In the Vienna experiments [13,14,17–19,23,24,74,75] measurements are performed by turning off the trapping potential at some time $t_0$, letting the gas expand freely and imaging the three-dimensional boson density after a time-of-flight $t_1$. The outcome of each such "single-shot" measurement is determined by the eigenvalues $e^{\frac{i}{2}\varphi_{a,s}(x,t)}$ of the bosonic vertex operators $e^{\frac{i}{2}\phi_{a,s}(x,t_0)}$ [26, 28]. As shown in [26], the result of a single measurement of the boson density after a time-of-flight $t_1$ in the regime relevant for the Vienna experiments can be well approximated by

$$\varrho_{\text{tof}}(x, \vec{r}, t_1, t_0) \simeq \rho_0 \left|f(\vec{r}, t_1)\right|^2 \times$$

$$\left|\int dx' \, G(x - x', t_1)\left[e^{i\frac{m}{2t_1}\vec{r}\cdot\vec{d}} e^{\frac{i}{2}(\varphi_s(x',t_0) + \varphi_a(x',t_0))} + e^{-i\frac{m}{2t_1}\vec{r}\cdot\vec{d}} e^{\frac{i}{2}(\varphi_s(x',t_0) - \varphi_a(x',t_0))}\right]\right|^2. \tag{14}$$

Here $\vec{d}$ is the distance between the minima of the double well, $x, x'$ and $\vec{r} = (y, z)$ respectively denote longitudinal and transverse coordinates, and $G(x, t)$ is the Green's function for a free particle

$$G(x, t) = \sqrt{\frac{m}{2\pi i t}} \exp\left(i\frac{m}{2t}x^2\right). \tag{15}$$

The function $f(\vec{r}, t)$ is an overall envelope whose precise from follows from the details of the trapping potential. By measuring $\varrho_{\text{tof}}$, the system collapses to a simultaneous eigenstate of all $e^{\frac{i}{2}\phi_{a,s}(x,t)}$. The outcome of such measurements can be simulated if one has access to distribution functions of the corresponding eigenvalues $e^{\frac{i}{2}\varphi_{a,s}(x,t)}$. Such distribution functions will be computed in Sec. 3.4. In principle, the observable (14) also contains small contributions from the density fields $\Pi_{a,s}(x)$ [26]. In order to treat these, the above description of a projective measurement has to be preceded by a diagonalization of the full observable, which now contains non-commuting fields. We do not pursue this further here because these effects are expected to be small in the regime where our low-energy approximation applies.

Experiments typically report results related to the quantity

$$R(x_0, \vec{r}, t_1, t_0) = \int_{x_0 - \ell/2}^{x_0 + \ell/2} dx\, \varrho_{\text{tof}}(x, \vec{r}, t_1, t_0) \tag{16}$$

$$= \rho_0 \left|f(\vec{r}, t_1)\right|^2 \int_{x_0 - \ell/2}^{x_0 + \ell/2} dx \left[|g_+(x)|^2 + |g_-(x)|^2 + 2\text{Re}\left(g_+(x)g_-^*(x)e^{i\frac{m\vec{r}\cdot\vec{d}}{t_1}}\right)\right],$$

where we have defined

$$g_\pm(x) = \int dx'\, G(x - x', t_1)e^{\frac{i}{2}\left(\varphi_s(x', t_0) \pm \varphi_a(x', t_0)\right)}. \tag{17}$$

## 2.3 Mode expansions for the two-component Luttinger liquid

The free boson Hamiltonians $H_{a,s}$ are diagonalized by the mode expansions (see e.g. [73])

$$\theta_j(x) = \theta_{j,0} + \frac{\pi x}{L}\delta N_j + i\sum_{q>0}\left(\frac{\pi K}{qL}\right)^{1/2}\sin qx\left(b_{j,q} - b_{j,q}^\dagger\right), \tag{18}$$

$$\phi_j(x) = \phi_{j,0} + \sum_{q>0}\left(\frac{\pi}{qKL}\right)^{1/2}\cos qx\left(b_{j,q} + b_{j,q}^\dagger\right), \tag{19}$$

where $q = \frac{\pi n}{L}$, $n \in \mathbb{Z}$, $\left[b_q, b_k^\dagger\right] = \delta_{q,k}$ and $\left[\delta N_j, \phi_{l,0}\right] = i\delta_{j,l}$. The zero modes $\delta N_j$ have integer eigenvalues. The Hamiltonians then take the form

$$H_j = \frac{v\pi}{2LK}\delta N_j^2 + \sum_{q>0} vq\, b_{j,q}^\dagger b_{j,q}, \quad j = a, s. \tag{20}$$

Going back to Eq. (5), we see that the hard-wall condition (4) is guaranteed by choosing the c-number $\theta_0$ such that

$$\theta(0) = \theta_0 \notin \mathbb{Z}. \tag{21}$$

It turns out to be useful in what follows to rewrite the mode-expansions in the form

$$\phi_l(x, t) = \sum_\nu u_\nu^{(l)}(x)\left(b_\nu(t) + b_\nu^\dagger(t)\right), \tag{22}$$

$$\partial_x\theta_l(x, t)/\pi = \sum_\nu w_\nu^{(l)}(x)\left(b_\nu(t) - b_\nu^\dagger(t)\right), \quad l = a, s. \tag{23}$$

Here we have introduced a multi-index $\nu = (l, q)$ that runs over all positive momenta $q \geq 0$ and the two sectors $l = a, s$ and we have defined

$$u^{(l)}_{(j,q)}(x) = \delta_{j,l} \begin{cases} \left(\frac{\pi}{qKL}\right)^{1/2} \cos qx, & \text{if } q \neq 0, \\ \frac{1}{2}\sqrt{\frac{1}{K}} & \text{if } q = 0, \end{cases} \tag{24}$$

$$w^{(l)}_{(j,q)}(x) = \delta_{j,l} \begin{cases} i\left(\frac{qK}{\pi L}\right)^{1/2} \cos qx, & \text{if } q \neq 0, \\ \frac{i}{L}\sqrt{K} & \text{if } q = 0, \end{cases} \tag{25}$$

$$b_{j,0} = \sqrt{K}\phi_{j,0} - \frac{i}{2}\sqrt{\frac{1}{K}}\delta N_j. \tag{26}$$

## 3  Self-consistent time-dependent harmonic approximation

Our aim is to determine the non-equilibrium evolution after a *quantum quench*: the system is prepared in a density matrix $\rho(0)$ that does not commute with the Hamiltonian (13). We moreover take the density matrix to be Gaussian for simplicity. The ensuing time evolution is described in the Schrödinger picture via the time evolving density matrix

$$\rho(t) = e^{-iHt}\rho(0)e^{iHt}. \tag{27}$$

As our Hamiltonian of interest (13) is not solvable we resort to an analysis by means of a SCTDHA [43, 69, 76–78]. Below we generalize the analysis of [69] to include the nonlinear interaction between the symmetric and antisymmetric sectors. The SCTDHA amounts to replacing the exact time evolution operator with

$$e^{-iHt} \longrightarrow U_{\text{SCH}}(t) = Te^{-i\int_0^t H_{\text{SCH}}(\tau)d\tau}, \tag{28}$$

where

$$H_{\text{SCH}}(t) = H_a + H_s + \int dx \Big[ f(x,t) + \phi_a(x)g^{(1)}(x,t) \\ + \Pi_s(x)g^{(2)}(x,t) + \phi_a^2(x)h^{(1)}(x,t) + \phi_a(x)\Pi_s(x)h^{(2)}(x,t) \Big]. \tag{29}$$

Here the functions $g^{(1,2)}(x,t)$ and $h^{(1,2)}(x,t)$ are determined self-consistently. In order to derive (29) we decompose the fields into their space and time dependent expectation values and their fluctuations

$$\phi_l(x,t) = \langle \phi_l(x,t) \rangle + \chi_l(x,t), \tag{30}$$
$$\Pi_l(x,t) = \langle \Pi_l(x,t) \rangle + \pi_l(x,t), \quad l = a, s. \tag{31}$$

Substituting this decomposition into the interaction part of the Hamiltonian (11) gives

$$H_\perp = -2t_\perp \int_0^L dx \left[\rho_0 + \sigma \langle \Pi_s \rangle + \sigma \pi_s\right]\left[\cos\langle\phi_a\rangle\cos\chi_a - \sin\langle\phi_a\rangle\sin\chi_a\right]. \tag{32}$$

In the next step we expand the Hamiltonian to quadratic order in fluctuations following [69], which gives

$$H_\perp \approx -2t_\perp \int dx \left[ \left(\rho_0 + \sigma \pi_s - \frac{1}{2}(\rho_0 + \sigma \langle \Pi_s \rangle)\chi_a^2 - \sigma \langle \chi_a \pi_s \rangle \chi_a\right)\cos\langle\phi_a\rangle \tag{33}$$

$$- \left((\rho_0 + \sigma(\pi_s + \langle \Pi_s \rangle))\chi_a - \frac{\sigma}{2}\langle \chi_a \pi_s \rangle \chi_a^2\right)\sin\langle\phi_a\rangle \right] e^{-\frac{1}{2}\langle\chi_a^2\rangle} + \text{const.}$$

After re-expressing this in terms of the original fields $\phi_a$ and $\Pi_s$, we arrive at Eq. (29), where the functions $h^{(j)}(x,t)$ and $g^{(j)}(x,t)$ are determined self-consistently by

$$
\begin{aligned}
h^{(1)}(x,t) &= \mathrm{Re}\overline{F}(x,t)/2, \\
h^{(2)}(x,t) &= \sigma\,\mathrm{Im}F(x,t), \\
g^{(1)}(x,t) &= \mathrm{Im}\overline{F}(x,t) - 2\left\langle \phi_a(x,t)\right\rangle h^{(1)}(x,t) - \left\langle \Pi_s(x,t)\right\rangle h^{(2)}(x,t), \\
g^{(2)}(x,t) &= -\sigma\,\mathrm{Re}F(x,t) - \left\langle \phi_a(x,t)\right\rangle h^{(2)}(x,t).
\end{aligned}
\tag{34}
$$

Here we have defined two functions

$$
\begin{aligned}
F(x,t) &= 2t_\perp \mathrm{Tr}\left[ U_{\mathrm{SCH}}(t)\rho(0)U_{\mathrm{SCH}}^\dagger(t)e^{i\phi_a(x)}\right], \\
\overline{F}(x,t) &= 2t_\perp \mathrm{Tr}\left[ U_{\mathrm{SCH}}(t)\rho(0)U_{\mathrm{SCH}}^\dagger(t)e^{i\phi_a(x)}\left(\rho_0 + \sigma\Pi_s(x)\right)\right].
\end{aligned}
\tag{35}
$$

One subtlety associated with the SCTDHA concerns the zero mode $\phi_{a,0}$. The spectrum of $\phi_{a,0}$ originally reflected the compact nature of the phase field $\phi_a(x) = \phi_a(x) + 2\pi$. The latter feature is lost in the SCTDHA, where fluctuations are assumed to be small but the fields themselves take arbitrary real values.

## 3.1 Gaussian initial states

In order to investigate the effects of the $\sigma$-term that couples the symmetric and antisymmetric sectors we want to start from a factorized state and study how correlations develop over time. An important requirement is related to our use of the SCTDHA: its accuracy strongly depends on the initial state obeying Wick's theorem. These two considerations lead us to consider the same class of initial states previously used in the literature [46–49]

$$
\rho(0) = \rho_a(0) \otimes \rho_s(0),
\tag{36}
$$

where $\rho_a(0) = |V,r,\varphi\rangle_{aa}\langle V,r,\varphi|$ is a Gaussian pure state

$$
|V,r,\varphi\rangle_a = \mathcal{N}\exp\left(\sum_{pq} V_p\left(\mathrm{sech}\,r^T\right)_{pq} b_{a,q}^\dagger + \sum_{p,q,k}\frac{1}{2}b_{a,p}^\dagger\left(\tanh r\right)_{pq}e^{i\varphi_{qk}}b_{a,k}^\dagger\right)|0\rangle_a.
\tag{37}
$$

It is useful to define new annihilation operators $\alpha_{a,k}$ satisfying

$$
\alpha_{a,k}|V,r,\varphi\rangle_a = 0,
\tag{38}
$$

which are related to the $b$-operators via the canonical transformation

$$
b_{a,q} = \sum_k (\cosh r)_{qk}\left[\alpha_{a,k} + V_k\right] + \left(\sinh r e^{i\varphi}\right)_{qk}\left[\alpha_{a,k}^\dagger + V_k^*\right].
\tag{39}
$$

In previous works it has been assumed that the symmetric sector is initialized in a thermal state [49]. We will follow this assumption, but in order to study the effects of spatial inhomogeneity we take our initial state to be given by a "displaced" thermal density matrix

$$
\rho_s = D(R)\frac{e^{-\beta H_s}}{\mathrm{Tr}\,e^{-\beta H_s}}D^\dagger(R),
\tag{40}
$$

where the displacement operators are defined via

$$
D^\dagger(R)b_{j,k}D(R) = b_{j,k} + R_{j,k}, \quad j = a,s.
\tag{41}
$$

This suggests the definition of displaced annihilation operators $\alpha_{s,k}$ via a constant shift

$$b_{s,k} = \alpha_{s,k} + R_{s,k}, \tag{42}$$

so that

$$\langle \alpha_{s,k} \rangle = 0, \tag{43}$$

on the initial state. Since $\rho_s(0)$ satisfies Wick's theorem, it is then completely fixed by the vector $R_{s,k}$ along with connected two-point functions of the fields. Using the mode expansion of $H_s$ from Eq. (20) we simply find bosonic occupation numbers for $q > 0$,

$$\left\langle b_{s,q}^\dagger b_{s,k} \right\rangle_c = \frac{\delta_{q,k}}{e^{\beta v q} - 1} \equiv n_{(s,q)}, \tag{44}$$

the anomalous expectation values $\langle b_{s,q} b_{s,q'} \rangle_c$ being zero. For the zero mode, the only expectation values on $\rho_s(0)$ that we will need are

$$\langle \delta N_s^2 \rangle_c = \frac{\sum_n e^{-\beta \frac{v\pi}{2KL} n^2} n^2}{\sum_n e^{-\beta \frac{v\pi}{2KL} n^2}}, \quad \langle \delta N_s \rangle = 0, \tag{45}$$

where the second identity implies $\mathrm{Im} R_{s,0}(0) = 0$. As will become clear in the next section, expectation values involving the field $\phi_{s,0}$ will never be required for the computation of physical quantities.

## 3.2 Equations of motion

The SCTDHA allows for a closed-form expression of the equations of motion. We will work in the Heisenberg picture from here onwards. The SCTDHA guarantees that time evolving annihilation operators can always be written as

$$b_\nu(t) = R_\nu(t) + S_{\nu\mu}(t)\alpha_\mu + T_{\nu\mu}^*(t)\alpha_\mu^\dagger, \tag{46}$$

where $\alpha_\mu$ are a set of bosonic creation and annihilation operators. We choose these to be given by

$$\alpha_\nu = \begin{cases} \alpha_{a,k} & \text{if } \nu = (a,k) \\ \alpha_{s,k} & \text{if } \nu = (s,k), \end{cases} \tag{47}$$

where the $\alpha_{a,k}$ are defined in (39) and the $\alpha_{s,k}$ in (42). For (46) to be a canonical transformation we require

$$SS^\dagger - T^*T^T = \mathbb{1}, \qquad ST^\dagger - T^*S^T = 0. \tag{48}$$

The initial conditions on $R, S$ and $T$ are given by

$$R_\mu(0) = \begin{cases} \sum_q (\cosh r)_{pq} V_q + (\sinh r e^{i\varphi})_{pq} V_q^* & \text{if } \mu = (a,p), \\ 0 & \text{else}, \end{cases}$$

$$S_{\nu,\mu}(0) = \begin{cases} (\cosh r)_{pq} & \text{if } \nu = (a,p), \ \mu = (a,q), \\ \delta_{pq} & \text{if } \nu = (s,p), \ \mu = (s,q), \\ 0 & \text{else}, \end{cases} \tag{49}$$

$$T_{\nu,\mu}^*(0) = \begin{cases} (\sinh r e^{i\varphi})_{pq} & \text{if } \nu = (a,p), \ \mu = (a,q), \\ 0 & \text{else}. \end{cases}$$

We note that the $\alpha_\mu$'s satisfy Wick's theorem on the initial state, along with $\langle \alpha_\mu \rangle = 0$ for all $\mu$.

The time evolution of any operator is then encoded in the time-dependence of the tensors $R, S$ and $T$, which we will now determine. To this end, we write the SCTDHA Hamiltonian in the generic form

$$H_{\text{SCH}}(t) = b_\nu^\dagger A_{\nu\mu}(t) b_\mu + \frac{1}{2} \left( b_\nu^\dagger B_{\nu\mu}^\dagger(t) b_\mu^\dagger + b_\nu B_{\nu\mu}(t) b_\mu \right)$$
$$+ C(t) + D_\nu(t) \left( b_\nu + b_\nu^\dagger \right) + E_\nu(t) \left( b_\nu - b_\nu^\dagger \right). \tag{50}$$

The matrices $A, B$ and vectors $D, E$ depend on the self-consistency functions $g^{(1,2)}$ and $h^{(1,2)}$, *cf.* Eqs. (34), and are given in Appendix A. Inserting the expansion (46) into the Heisenberg equation of motion,

$$i \frac{d}{dt} b_\nu(t) = U_{\text{SCH}}(t) [b_\nu, H_{\text{SCH}}(t)] U_{\text{SCH}}^\dagger(t), \tag{51}$$

yields a system of coupled, first order differential equations

$$i \dot{R}_\nu(t) = A_{\nu\mu}(t) R_\mu(t) + B_{\nu\mu}^\dagger(t) R_\mu^*(t) + D_\nu(t) - E_\nu(t)$$
$$i \dot{S}_{\nu\mu}(t) = A_{\nu\lambda}(t) S_{\lambda\mu}(t) + B_{\nu\lambda}^\dagger(t) T_{\lambda\mu}(t) \tag{52}$$
$$-i \dot{T}_{\nu\mu}(t) = A_{\nu\lambda}^*(t) T_{\lambda\mu}(t) + B_{\nu\lambda}^T(t) S_{\lambda\mu}(t).$$

This system of ODE's is *nonlinear*: as a result of the self-consistency functions (34) on which the tensors $A, B, D$ and $E$ depend, these tensors are themselves functions of $R, S$ and $T$, which therefore enter the system (52) in nonlinear combinations. To simplify some of the following equations we introduce linear combinations

$$Q_{\nu\mu}(t) = S_{\nu\mu}(t) + T_{\nu\mu}(t), \quad \overline{Q}_{\nu\mu}(t) = S_{\nu\mu}(t) - T_{\nu\mu}(t). \tag{53}$$

In terms of these functions mode expansions of the time evolved fields take the form

$$\phi_a(x,t) = \sum_\nu u_\nu^{(a)}(x) \left( 2\text{Re} R_\nu(t) + \sum_\mu \left[ Q_{\nu\mu}(t) \alpha_\mu + Q_{\nu\mu}^*(t) \alpha_\mu^\dagger \right] \right), \tag{54}$$

$$\Pi_l(x,t) = \sum_\nu w_\nu^{(l)}(x) \left( 2i\text{Im} R_\nu(t) + \sum_\mu \left[ \overline{Q}_{\nu\mu}(t) \alpha_\mu - \overline{Q}_{\nu\mu}^*(t) \alpha_\mu^\dagger \right] \right). \tag{55}$$

The functions (35) can then be computed using Wick's theorem for the $\alpha$-operators, based on the above expressions. This closes the system of ODE's (52). The zero mode in the symmetric sector $\phi_{s,0}$ reflects the compact nature of the phase field $\phi_s$ and therefore needs to be treated separately from the finite momentum modes. We therefore define a field

$$\widetilde{\phi_s}(x) \equiv \phi_s(x) - \phi_{s,0}, \tag{56}$$

which time evolves as

$$\widetilde{\phi_s}(x,t) = \sum_{\nu \neq (s,0)} u_\nu^{(s)}(x) \left( 2\text{Re} R_\nu(t) + \sum_\mu \left[ Q_{\nu\mu}(t) \alpha_\mu + Q_{\nu\mu}^*(t) \alpha_\mu^\dagger \right] \right). \tag{57}$$

Importantly the zero mode $\phi_{s,0}$ does not get generated under Heisenberg time evolution of other fields. This is easily checked by inspection of the Hamiltonian (13) which is seen to not involve $\phi_{s,0}$. This in turn implies that the zero mode cannot appear on the rhs of the Heisenberg equation of motion (51). Since we can express the zero mode at $t = 0$ as

$$\phi_{s,0} = \left( \alpha_{(s,0)} + \alpha_{(s,0)}^\dagger \right) / \sqrt{4K}, \tag{58}$$

we conclude that this linear combination of $\alpha$-operators does not appear in the sums over modes in (54,55) except in the expansion for $\phi_s(x,t)$, where it occurs in the term with $\nu = (s,0)$. This directly leads to

$$\operatorname{Re} Q_{\nu,(s,0)}(t) = 0 \quad \forall\ \nu \neq (s,0), \qquad \operatorname{Im} \overline{Q}_{\nu,(s,0)}(t) = 0 \quad \forall\ \nu. \tag{59}$$

### 3.3 Self-consistent expectation values

#### 3.3.1 One-point functions

As all relevant one-point functions of $\alpha_\nu$ and $\delta N_s$ are zero we have

$$\left\langle \widetilde{\phi}_s(x,t) \right\rangle = 2 \sum_{\nu \neq (s,0)} u_\nu^{(s)}(x) \operatorname{Re} R_\nu(t), \tag{60}$$

$$\left\langle \phi_a(x,t) \right\rangle = 2 \sum_\nu u_\nu^{(a)}(x) \operatorname{Re} R_\nu(t), \tag{61}$$

$$\left\langle \Pi_l(x,t) \right\rangle = 2i \sum_\nu w_\nu^{(l)}(x) \operatorname{Im} R_\nu(t). \tag{62}$$

#### 3.3.2 Two-point functions

Comparing the definitions from Section 3.1 to the initial conditions (49), we find that for any $\nu, \mu$,

$$\mathfrak{g}_{\nu,\mu} = \left\langle \alpha_\nu^\dagger \alpha_\mu \right\rangle = \left\langle \alpha_\nu \alpha_\mu^\dagger \right\rangle - \delta_{\nu,\mu} = \delta_{\nu,\mu} \begin{cases} 0 & \text{if } \nu \in \{(a,q),(s,0)\} \\ n_{(s,q)} & \text{if } \nu \in \{(s,q)|q \neq 0\}. \end{cases} \tag{63}$$

If we define $P_0^{(s)}$ to be the projector on the symmetric zero modes, along with its complement $\tilde{\mathbb{1}} = \mathbb{1} - P_0^{(s)}$, we then find the following connected two-point functions

$$\left\langle \phi_j(x,t)\phi_l(y,t) \right\rangle_c = u^{(j)}(x)\left( 2\operatorname{Re}(Q^* \mathfrak{g} Q^T) + Q\tilde{\mathbb{1}}Q^\dagger + \frac{\langle \delta N_{s0}^2 \rangle}{K} \operatorname{Im} Q P_0^{(s)} \operatorname{Im} Q^T \right) u^{(l)}(y),$$

$$\left\langle \phi_j(x,t)\Pi_l(y,t) \right\rangle_c = -u^{(j)}(x)\left( 2i\operatorname{Im}(Q\mathfrak{g}\overline{Q}^\dagger) + Q\tilde{\mathbb{1}}\overline{Q}^\dagger + i\frac{\langle \delta N_{s0}^2 \rangle}{K} \operatorname{Im} Q P_0^{(s)} \operatorname{Re} \overline{Q}^T \right) w^{(l)}(y). \tag{64}$$

In the above, indices on all matrices and vectors have been suppressed for conciseness. If we want to consider the field $\widetilde{\phi}_s$ instead of $\phi_s$, we need leave out the symmetric zero mode term. This leads, for instance, to

$$\left\langle \widetilde{\phi}_s(x,t)\Pi_l(y,t) \right\rangle_c = u^{(j)}(x)\left( P_0^{(s)} - \mathbb{1} \right) \times$$

$$\times \left( 2i\operatorname{Im}(Q\mathfrak{g}\overline{Q}^\dagger) + Q\tilde{\mathbb{1}}\overline{Q}^\dagger + i\frac{\langle \delta N_{s0}^2 \rangle}{K} \operatorname{Im} Q P_0^{(s)} \operatorname{Re} \overline{Q}^T \right) w^{(l)}(y), \tag{65}$$

and analogous modifications for $\left\langle \widetilde{\phi}_s(x,t)\widetilde{\phi}_s(y,t) \right\rangle_c$ and $\left\langle \widetilde{\phi}_s(x,t)\phi_a(y,t) \right\rangle_c$.

### 3.4 Full distribution functions

Individual measurement outcomes in interference experiments of interest [24] are fully determined by the eigenvalues $\varphi_a$ and $\widetilde{\varphi}_s$ of the phase fields $\phi_a$ and $\widetilde{\phi}_s$ [26], *cf.* Eq. (14). To model the outcomes of such measurements we therefore require the time-dependent distribution

functions for $\varphi_a$ and $\widetilde{\varphi_s}$. These can be determined in the framework of the SCTDHA [43, 69]. For the case at hand, we first expand the eigenvalues of the phase fields as Fourier series,

$$\widetilde{\varphi_s}(x,t) = \sum_{\mu \neq (s,0)} u_\mu^{(s)}(x) f_{\mu,t}, \qquad \varphi_a(x,t) = \sum_\mu u_\mu^{(a)}(x) f_{\mu,t}. \tag{66}$$

Here we have again used our multi-index notations $\mu = (j, q)$, where $j = a, s$ labels the sector and $q$ the momentum. Each measurement selects a particular set of Fourier coefficients and we denote the averages over many measurements by

$$\overline{f_{\mu,t}}, \quad \overline{f_{\mu,t}\, f_{\nu,t}} \quad \text{etc.} \tag{67}$$

The mean values for the Fourier coefficients can be read off from the one-point functions calculated earlier, *cf.* Eqs. (60,61)

$$\overline{f_{\mu,t}} = 2\operatorname{Re} R_\mu(t). \tag{68}$$

The object of interest is then the time-dependent joint probability distribution $P$ of Fourier coefficients $\{f_\mu\}$. Within the SCTDHA all cumulants of $\phi_{a,s}$ other than the variance vanish, so that this probability distribution is Gaussian

$$P(\{f_\mu\}, t) = \frac{1}{(2\pi)^{N/2}} \frac{1}{\sqrt{\det M(t)}} \exp\left(-\frac{1}{2} \sum_{\mu,\nu} \left(f_\mu - \overline{f_{\mu,t}}\right) M_{\mu\nu}^{-1}(t) \left(f_\nu - \overline{f_{\nu,t}}\right)\right). \tag{69}$$

Here $N$ is the total number of Fourier modes retained in (66). Noting that

$$\left\langle \phi_j(x,t)\phi_l(y,t)\right\rangle_c = u_\mu^{(j)}(x)\left(\overline{f_{\mu,t}f_{\nu,t}} - \overline{f_{\mu,t}}\,\overline{f_{\nu,t}}\right) u_\nu^{(l)}(y), \quad j, l \in \{a, s\} \tag{70}$$

and comparing to Eq. (64), we can directly read off the covariance matrix as well:

$$M(t) = 2\operatorname{Re}(Q\mathfrak{g}Q^\dagger) + QQ^\dagger + \frac{\langle \delta N_{s0}^2\rangle}{K} \operatorname{Im}Q P_0^{(s)} \operatorname{Im}Q^T. \tag{71}$$

Having obtained a time-dependent probability distribution for the coefficients $\{f_{\mu,t}\}$, we can directly model experiments: we draw coefficients $\{f_{\mu,t}\}$ from the distribution (69), reconstruct the corresponding eigenvalues (66), and insert these in the time-of-flight density (14) to compute the measured density profile. We note that in the notations used above the set $\{f_\mu\}$ contains the non-physical Fourier coefficient $f_{(s,0)}$. This quantity does not enter the observable (14), and can simply be discarded, whenever a set of coefficients is drawn from $P\left(\{f_\mu\}, t\right)$.

By repeating the above procedure for modelling a measurement many times over we can reconstruct the full distribution function of any observable that depends only on the phase fields $\phi_{a,s}$. In what follows, we will focus on the "interference term" in the spatially integrated density after time-of-flight $R_{\text{tof}}(x_0, \vec{r}, t_1, t_0)$ defined in (16). The eigenvalues of this observable are proportional to

$$\mathcal{I}_\ell\left(\{f_\mu\}, x_0, t_0, t_1\right) = \frac{1}{\ell} \int_{x_0-\ell/2}^{x_0+\ell/2} dx\, g_+(x) g_-^*(x), \tag{72}$$

where $g_\pm(x)$ are defined in (17) and are related to the coefficients $f_\mu$ via (66). Motivated by the experimental data analyses of Refs [21, 22, 68] we parametrize the interference term (72) as

$$\mathcal{I}_\ell\left(\{f_\mu\}, x_0, t_0, t_1\right) = C_\ell(x_0, t_0, t_1, \{f_\mu\}) e^{i\Phi_\ell(x_0, t_0, t_1, \{f_\mu\})}. \tag{73}$$

By drawing many sets $\{\mathfrak{f}_\mu\}$ of coefficients from the distribution function $P\left(\{\mathfrak{f}_\mu\}, t\right)$ and plotting the resulting values of $\Phi_\ell$ or $C_\ell$ in a normalized histogram, we converge to probability distributions $P_{\Phi_\ell, C_\ell}$ for these quantities. These distribution functions can formally be written as

$$P_{\Phi_\ell}(\alpha, t_0, t_1) = \left(\prod_\mu \int d\mathfrak{f}_\mu\right) \delta\left(\alpha - \operatorname{Arg} \mathcal{I}_\ell\left(\{\mathfrak{f}_\mu\}, x_0, t_0, t_1\right)\right) P\left(\{\mathfrak{f}_\mu\}, t_0\right), \tag{74}$$

$$P_{C_\ell}(\gamma, t_0, t_1) = \left(\prod_\mu \int d\mathfrak{f}_\mu\right) \delta\left(\gamma - \operatorname{Abs} \mathcal{I}_\ell\left(\{\mathfrak{f}_\mu\}, x_0, t_0, t_1\right)\right) P\left(\{\mathfrak{f}_\mu\}, t_0\right). \tag{75}$$

## 4 Results for experimentally relevant initial states

### 4.1 Choice of initial state

We now specialize to an initial state that is often used in the literature, see e.g. [46–49]. In [46], a quasi-classical argument is used to conjecture how the state of a pair of elongated Bose gases follows from the splitting process of a single gas. It is reasoned that when splitting a gas, each particle has an equal probability to end up in well 1 or in well 2. The relative particle number resulting from this Poisson process is thus a stochastic variable with mean zero and variance proportional to the particle density. Assuming short-range correlations, one arrives at

$$\langle \Pi_a(x, 0)\Pi_a(y, 0)\rangle_c = \frac{\eta \rho_0}{2} \delta_\xi(x - y), \tag{76}$$

with $\eta$ a phenomenological parameter which we will set to 1. Following [49], the delta function above is understood as a flat sum over plane waves running up to momentum $\pi/\xi$. To reproduce this initial two-point function, it suffices to use the initial state (37), with $r$ a real and diagonal matrix and $\varphi = 0$. The resulting initial condition on $\overline{Q}$,

$$\overline{Q}(0)_{(a,j)(a,k)} = \delta_{jk} e^{-r_{jj}}, \tag{77}$$

then leads to

$$\langle \Pi_a(x, 0)\Pi_a(y, 0)\rangle_c = \frac{K}{L^2} e^{-2r_{00}} + \sum_{j>0} \frac{qK}{\pi L} \cos\left(q_j x\right) \cos\left(q_j y\right) e^{-2r_{jj}}. \tag{78}$$

Comparing Eqs. (76) and (49), we can thus read off

$$e^{-2r_{jk}} = \delta_{jk} \begin{cases} \frac{L\eta\rho_0}{2K} & \text{if } q = 0, \\ \frac{\pi\eta\rho_0}{qK} & \text{if } q > 0, \end{cases} \tag{79}$$

for the antisymmetric sector.

For the symmetric sector, we again follow Refs. [47–49]: the above quasi-classical splitting argument applies to the relative degrees of freedom, leaving the symmetric combinations of densities and phases unaltered. In [49], the symmetric sector is therefore taken to be in a finite temperature equilibrium state. We adhere to this conjecture here and use the thermal density matrix described in Section 3.1, thereby fixing the initial conditions for both $T$ and $S$ in conjunction with the above discussion. Finally, the initial conditions for $R$ can be used to enforce various initial profiles on the density and phase fields in both sectors, which we will explore in Sec. 4.3 below.

## 4.2 Experimental parameters

We fix the parameters for our plots by following Ref. [21]: the one-dimensional density is taken to be $\rho_0 = 45\,\mu\text{m}^{-1}$, the coherence length is $\xi = \hbar\pi/mv = \pi \times 0.42\,\mu\text{m}$, the sound velocity is given by $v \approx 1.738 \cdot 10^{-3}\,\text{m/s}$ and the Luttinger parameter in our conventions is $K \approx 28$. We take the one-dimensional box size as large as we can achieve for a given value of the cutoff length scale, which amounts to $L = 80\,\mu\text{m}$. This is comparable to the size reported in [21]. In all figures, time is measured in units of the *traversal time* [4], $t_{\text{tr}} = L/2v$, which is the time it takes for a light cone to reach the edge of the system from the centre of the box. We work at a temperature of $5\,\text{nK}$ throughout. This ensures that we are well in the scaling regime, at an energy density that is $1/8$ times the cutoff energy $\epsilon_c = v\pi/\xi$, with $\xi$ the coherence length. We have chosen $t_\perp = 15\,\text{Hz}$, which guarantees that the gap is of the same order as the temperature for the above parameters. The only exception to this choice is Fig. 2, where we take $t_\perp \approx 1.17\,\text{Hz}$ following Ref. [69], to enable a comparison with the case of periodic boundary conditions as presented in that paper.

## 4.3 Time evolution

We now consider time evolution under the SCTDHA Hamiltonian (29), with the initial condition described in Sec. 4.1. Throughout, we choose $R(0)$ such that

$$\langle\phi_a(x,0)\rangle = 0.2, \quad \langle\Pi_a(x,0)\rangle = 0. \tag{80}$$

The one-point functions $\left\langle\widetilde{\phi_s}(x,0)\right\rangle$ and $\langle\Pi_s(x,0)\rangle$ will be given different spatial profiles, to investigate the effects of broken translational invariance.

### 4.3.1 No coupling between symmetric and antisymmetric sectors ($\sigma = 0$)

We will start from the situation where

$$\left\langle\widetilde{\phi_s}(x,0)\right\rangle = 0 = \langle\Pi_s(x,0)\rangle \tag{81}$$

and $\sigma = 0$. This will serve as our benchmark, as it most closely resembles the translationally invariant scenario described in [69] in which the (anti)symmetric sectors remain uncorrelated. It is characterized by Josephson oscillations between density and phase, see Fig. 1(a), with a phase variance that initially grows, and then shows oscillating behavior, see Fig. 1(b).

To connect with our previous work [69] we include a comparison between results from that paper, where periodic boundary conditions were used, and the results derived for a box geometry in the present paper. Fig. 2 shows that the two geometries give extremely similar results in the centre of the trap for times below the traversal time, whereas deviations do occur after this time. It should also be noted that in [69] and Fig. 2, results are presented for smaller tunnel couplings ($t_\perp \approx 1.17\,\text{Hz}$) than in the rest of this paper. The reason for choosing these values in [69] was that for a relatively shallow field potential, the inharmonicity of the cosine in the sine-Gordon model manifests itself more strongly, making deviations from the purely quadratic theory more apparent. For the purposes of this paper, however, it is more interesting to look at relatively large tunnel-couplings ($t_\perp = 15\,\text{Hz}$, see Sec. 4.2), as this enhances the coupling between the sectors in which we are interested.

### 4.3.2 Finite coupling between sectors ($\sigma > 0$) and homogeneous initial conditions

We next investigate different values of the coupling constant $\sigma$, and the resulting mixing between the sectors. Fig. 3 shows results for $\sigma = 0, 1/2, 1, 3/2, 2$, starting from completely flat profiles, as in Eqs. (80), (81). When increasing $\sigma$, the phase oscillations remain essentially

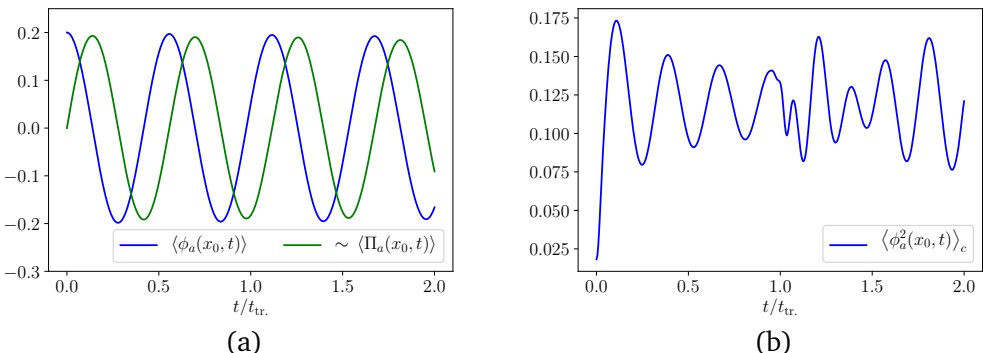

Figure 1: (a) Josephson oscillations of relative density (arbitrary units) and phase (radians) at the centre of the gas, $x_0 = L/2$. (b) initial growth and oscillations of the variance of the relative phase. The initial phase and density profiles are chosen according to Eqs. (80,81) and coupling between the sectors is absent in these pictures, meaning $\sigma = 0$.

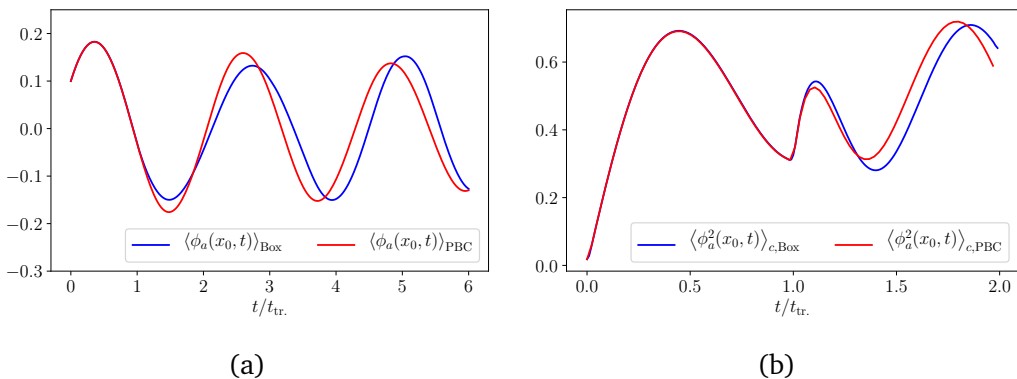

Figure 2: Comparison between results for box boundary conditions (blue) and periodic boundary conditions (red). The curves are in perfect agreement until the traversal time $t_{\rm tr} = L/2v$, after which deviations occur. (a) Josephson oscillations of phase (radians) at the centre of the gas, $x_0 = L/2$. (b) initial growth and subsequent oscillations in the variance of the relative phase.

unchanged. A stronger effect is visible in the covariance between $\phi_a$ and $\widetilde{\phi_s}$, however. To quantify this, we define

$$C(x,t) \equiv \frac{\left\langle \widetilde{\phi_s}(x,t)\phi_a(x,t) \right\rangle_c}{\sqrt{\left\langle \widetilde{\phi_s}(x,t)\widetilde{\phi_s}(x,t) \right\rangle_c \left\langle \phi_a(x,t)\phi_a(x,t) \right\rangle_c}}. \tag{82}$$

As can be seen in Fig. 3(b), the covariance $C(x,t)$ increases to appreciable values as $\sigma$ is increased. We also note that for larger values of $\sigma$, the variance of the relative phase increases somewhat for times below the traversal time, see Fig. 4.

It is also instructive to consider the energy flow between different terms in the Hamiltonian.

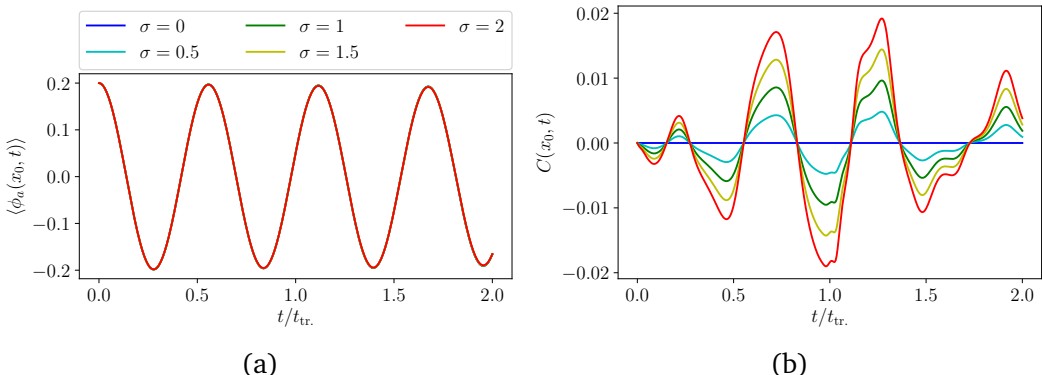

Figure 3: (a) time evolution of the phase in the antisymmetric sector at the box centre $x_0 = L/2$. Curves are displayed for different values of $\sigma$, with a flat initial density profile $\langle \Pi_s(x) \rangle = 0$. A change of $\sigma$ has no appreciable effect on this observable. (b) a somewhat stronger effect is the development of correlations between $\phi_{a,s}$, where the normalized covariance from Eq. (82) is displayed, for $x_0 = L/2$.

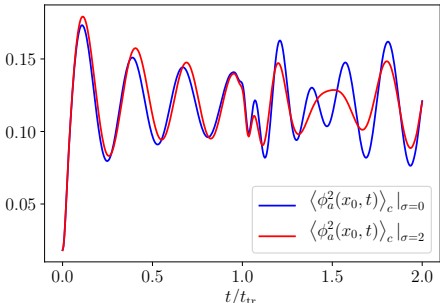

Figure 4: Variance of the relative phase, for $\sigma = 0$ (blue) and $\sigma = 2$ (red). A slight increase in the variance is visible for the larger value of $\sigma$ for times below $t_{\text{tr}} = L/2v$.

To this end we define the following quantities

$$e_{a,0}(t) = \frac{\langle H_a \rangle}{L}, \quad e_{a,\perp}(t) = -\frac{2t_\perp \rho_0}{L} \int_0^L dx \, \langle \cos \phi_a(x) \rangle, \quad e_{sG}(t) = e_{a,0}(t) + e_{a,\perp}(t),$$

$$e_{\text{int}}(t) = -\frac{2t_\perp \sigma}{L} \int_0^L dx \, \langle \Pi_s(x) \cos \phi_a(x) \rangle, \quad e_s(t) = e_{\text{int}}(t) + \langle H_s(t) \rangle / L. \tag{83}$$

We note that the total energy density, which is given by $e_{sG}(t) + e_{\text{int}}(t) + \langle H_s \rangle / L$, is independent of time, as required for a closed quantum system. Since we are interested in the time dependence of the various energy densities we subtract their values in the initial state and consider

$$\Delta e_j(t) \equiv e_j(t) - e_j(0). \tag{84}$$

To quantify the effects of the $\sigma$-coupling on the flow of energy from and to the sine-Gordon model we show $\Delta e_{SG}(t)$ in Fig. 5. To ascertain which fraction of the energy change is due to the kinetic and interaction parts of the sine-Gordon model we also show $\Delta e_a(t)$ and $\Delta e_{\perp,a}(t)$ in Fig. 5(a). We observe that the change in $\Delta e_{SG}(t)$ is very small, as significantly larger changes in $\Delta e_a(t)$ and $\Delta e_{\perp,a}(t)$ largely compensate each other. In Fig. 5(b) we show how much of the energy from the sine-Gordon model $\Delta e_{SG}(t)$ ends up in the new interaction term $e_{\text{int}}(t)$ and how much goes to $\langle H_s(t) \rangle / L$.

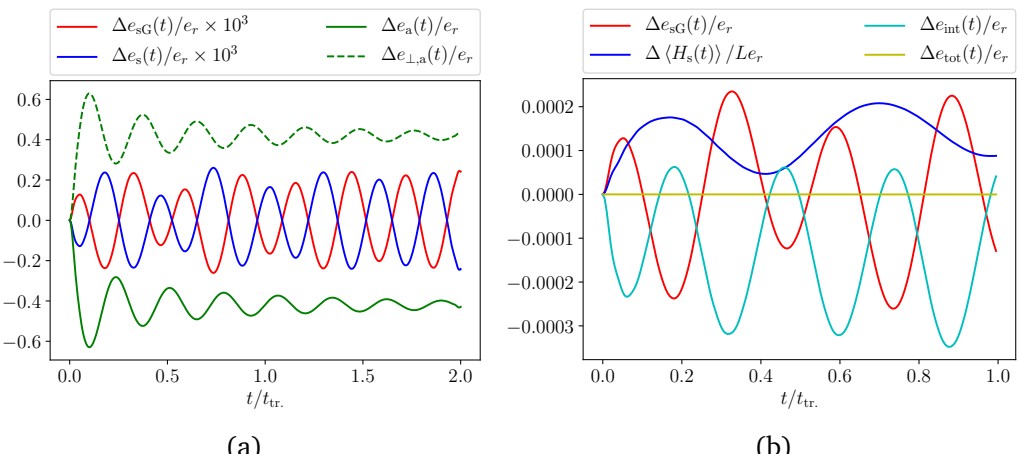

(a)                                                            (b)

Figure 5: Energy flow between the different terms in Eqs. (83), as a ratio with the reference scale $e_r = \langle H_s(0) \rangle / L$.

### 4.3.3 Finite coupling between sectors ($\sigma > 0$) and inhomogeneous initial conditions

As a next step, we investigate the effect of initial density profiles $\langle \Pi_s(x) \rangle$ that are spatially inhomogeneous. These profiles will evolve in time as is shown in Fig. 6 (a,b). The profiles

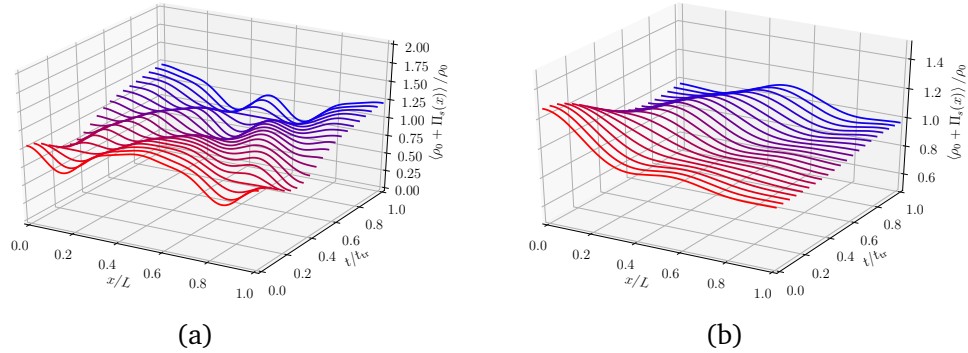

(a)                                                            (b)

Figure 6: Examples of the time evolution of the density profile for $\sigma = 0$. The initial profile in (a) is symmetric around the origin, while the one in (b) is not.

$\langle \phi_a(x) \rangle$ and $\langle \Pi_a(x) \rangle$ are strongly affected by the strength of the $\sigma$-coupling to the inhomogeneous profile $\langle \Pi_s(x) \rangle$ and develop inhomogeneities as a consequence. This is illustrated in Figs. 7(a,b) and has repercussions for the Josephson oscillations. The latter now display spatial variations, which are caused by the oscillation frequency becoming $\sigma$- and position-dependent due to the presence of the space-dependent $\Pi_s(x)$-field in the interaction term. This local and $\sigma$-dependent frequency is illustrated in Fig. 8. The spatial average of the phase, which is equal to the zero mode $\phi_{a0}$, does not show any $\sigma$-dependence in its Josephson frequency, see Figs. 9,10. In this case, however, a $\sigma$-dependent modulation in the amplitudes is visible: the oscillations at different points in the box move out of phase due to the spatially varying frequency mentioned above. This leads to a decrease in the spatial average. For an inhomogeneous profile of $\langle \Pi_s(x,0) \rangle$, the covariance grows in time, in resemblance with the homogeneous case. This happens to an extent that is roughly proportional to $\sigma$. The same can be said of the energy flow between the (anti)symmetric sectors, as shown in Fig. 11. The effects of the sector coupling term become stronger when we increase $\sigma$, but in the window of applicability of our bosonization approach the effects remain small.

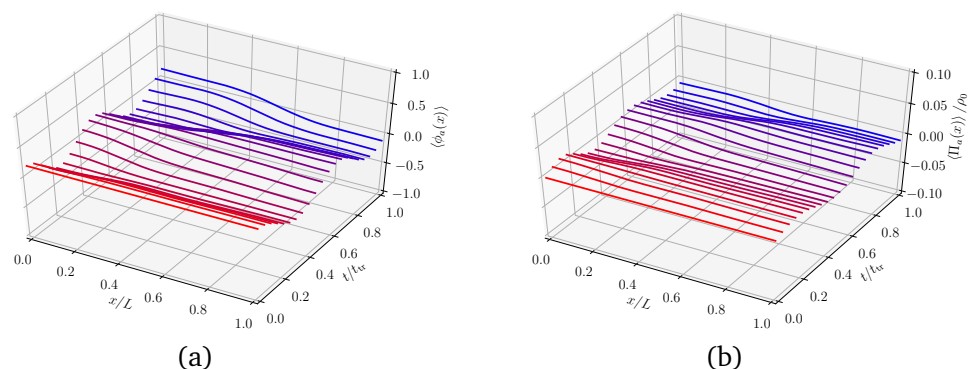

Figure 7: (a) The time and position dependence of $\langle\phi_a(x)\rangle$ corresponding to the same initial condition as Fig. 6(a) with $\sigma = 2$. We see that the initially flat profile develops inhomogeneities due to the sector coupling. (b) the same as panel (a), but showing $\langle\Pi_a(x)\rangle$.

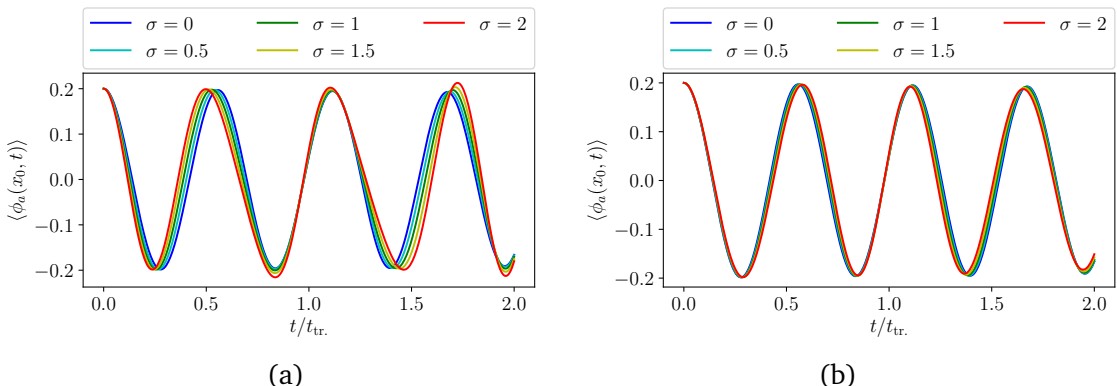

Figure 8: Time dependence of the relative phase in the centre of the box for the same initial conditions as Fig. 6(a) and (b), respectively.

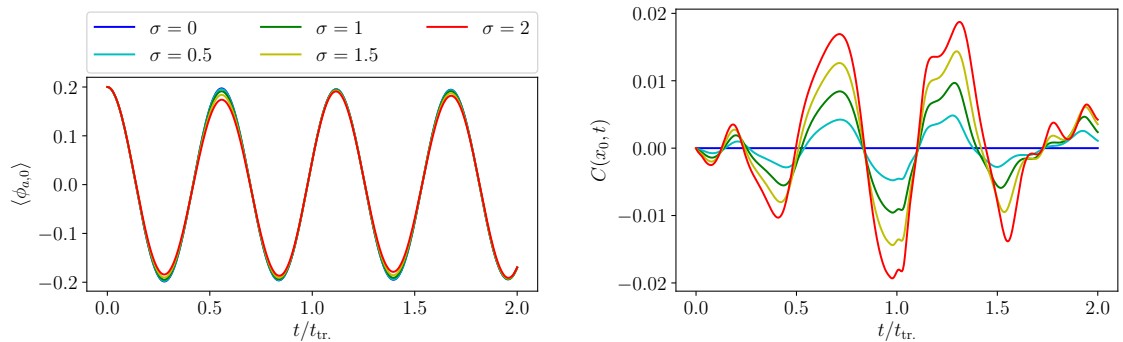

Figure 9: Time dependence of the space-averaged relative phase and the covariance $C(x_0, t)$ (82) in the centre of the box for the profiles shown in panel (a) of Fig. 6.

### 4.3.4 Distribution functions of the density after time of flight

As described in Sec. 3.4, our formalism allows the construction of distribution functions for the measured density after time-of-flight expansion. As a proof of principle we present such distribution functions in Fig. 12, for the observables $\Phi_\ell$ and $C_\ell$ defined in Eq. (73).

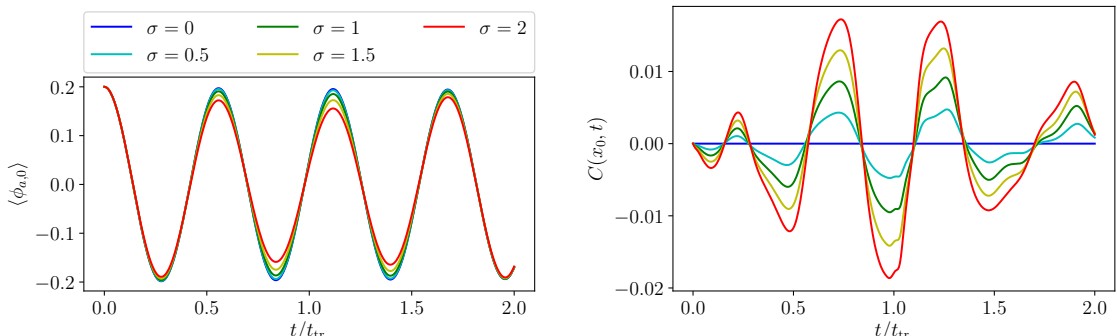

Figure 10: Time dependence of the space-averaged relative phase and the covariance $C(x_0, t)$ (82) in the centre of the box for the profiles shown in panel (b) of Fig. 6.

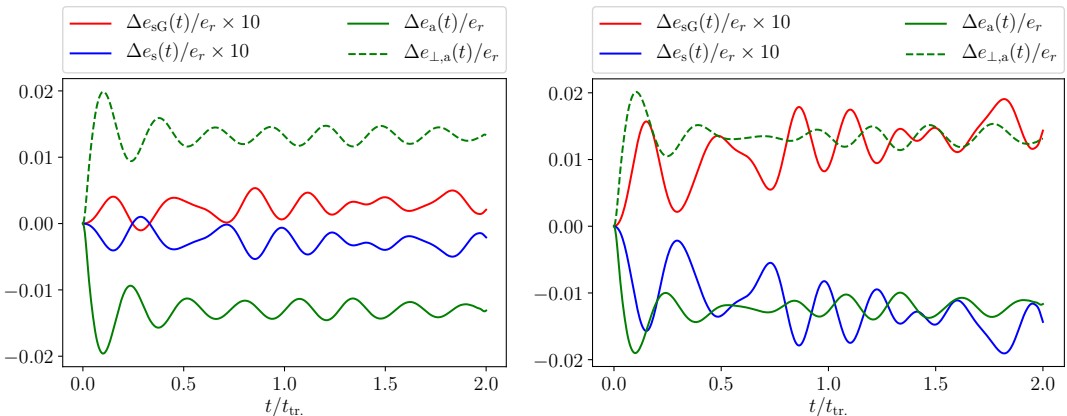

Figure 11: Energy flow between different terms in Eqs. (83), as a ratio with the reference scale $e_r = \langle H_s(0) \rangle / L$. Results are shown for the density profile from Fig. 6(a), with $\sigma = 1$ (left panel) and $\sigma = 2$ (right panel).

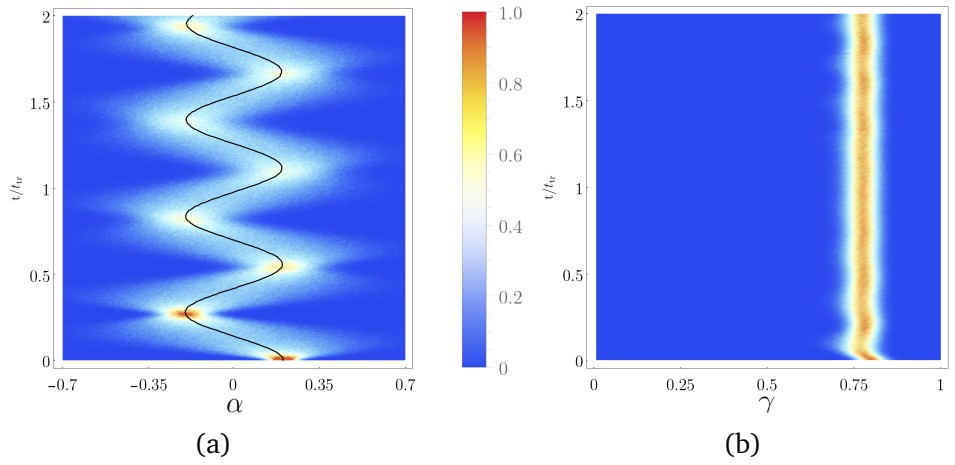

(a)  (b)

Figure 12: Distribution functions $P_{\Phi_\ell}(\alpha, t, t_1)$ (a) and $P_{C_\ell}(\gamma, t, t_1)$ for the observables $\Phi_\ell$ and $C_\ell$ defined in Eq. (73). We choose a time of flight $t_1 = 15\,\text{ms}$ and integration length $\ell = 20\,\mu\text{m}$. The density profile used for these plots is homogeneous, with $\sigma = 1$.

# 5 Conclusion

We have extended the theory for non-equilibrium dynamics in pairs of elongated, tunnel-coupled Bose gases using a self-consistent time-dependent harmonic approximation (SCTDHA) in the low-energy scaling limit (energy scales $\lesssim 5$ nK). In contrast to earlier works, we have studied the effect of a relevant perturbation which couples the (anti)symmetric sectors describing (anti)symmetric combinations of the two Bose gas phases. On top of this, we have dropped the assumption of translational invariance by placing the system in a box and by imposing inhomogeneous initial density profiles.

Starting from an initial state in which these sectors are uncorrelated, the coupling of the sectors under time evolution leads to a number of new but weak effects. First of all we observe the development of correlations between the sectors over time. This effect is present for all initial states we have considered, but the covariance between the sectors never reaches more than a few percent of the geometric mean of the variances. Second, the spreading of correlations is accompanied by a small transfer of energy between the sectors. And finally, the presence of the coupling term makes the dynamics in the antisymmetric sector susceptible to the breaking of translational invariance in the symmetric sector. The well-known Josephson oscillations of relative density and phase are modulated when taking an inhomogeneous initial density profile of the symmetric sector. This shows that the role of the trapping potential, which creates strong inhomogeneities, may play a more important role in experiment than was previously assumed. However, the model presented here does not capture the puzzling damping phenomenon observed recently [24, 66, 67]. This is not surprising given that our box potential is very different from the quadratic potential used in experiment. In future experiments, however, the box potential is likely to be used, which adds to the relevance of the calculations presented here.

We conclude that *(i)* the new term coupling the (anti)symmetric sectors leads to very weak effects. This means that the simulation of a sine-Gordon model using the setup described in this paper should not be severely hampered by the presence of this term. *(ii)* we have shown that it is possible to treat states with broken translational invariance in the SCTDHA formalism as presented in [69]. Combined with the sector coupling, we find that inhomogeneities in the density can have weak but nontrivial effects on the amplitude of Josephson oscillations. This means that the trapping potential is likely to have an effect on the dynamics probed in experiment. In a forthcoming paper, we will present a study of the projected Hamiltonian (3) in a microscopic analysis that takes a quadratic longitudinal potential into account. It would be interesting to combine such a microscopic approach with low-energy effective field theory calculations in the presence of such a quadratic trapping potential. This could be done by combining a SCTDHA with recent results [79, 80] for inhomogeneous Luttinger liquids [81–84]. However, the calculations using the box potential presented here may gain additional relevance when more experiments using a box potential, such as Refs. [85, 86], are performed.

## Acknowledgements

We are grateful to Jörg Schmiedmayer, Thomas Schweigler and Marine Pigneur for stimulating discussions and to the Erwin Schrödinger International Institute for Mathematics and Physics for hospitality and support during the programme on *Quantum Paths*. This work was supported by the EPSRC under grant EP/S020527/1 and YDvN is supported by the Merton College Buckee Scholarship and the VSB, Muller and Prins Bernhard Foundations.

# A Tensors occurring in $H_{\mathrm{SCH}}(t)$

The tensors occurring in $H_{\mathrm{SCH}}(t)$ as written in Eq. (50) are given by

$$
A = \left(
\begin{array}{cc|cc}
\ddots & \cdot\cdot & \ddots & \cdot\cdot \\
vq\,\delta_{q,k} + 2\Delta^{(1)}_{q,k}(t)\,u^{(a)}_{a,q}(0)u^{(a)}_{a,k}(0) & & \Delta^{(2)}_{q,k}(t)\,u^{(a)*}_{a,q}(0)w^{(s)}_{s,k}(0) & \\
\cdot\cdot & \ddots & \cdot\cdot & \ddots \\
\hline
\ddots & \cdot\cdot & \ddots & \cdot\cdot \\
\Delta^{(2)}_{q,k}(t)\,u^{(a)}_{a,k}(0)w^{(s)*}_{s,q}(0) & & vq\,\delta_{q,k} & \\
\cdot\cdot & \ddots & \cdot\cdot & \ddots
\end{array}
\right),
$$

$$
B = \left(
\begin{array}{cc|cc}
\ddots & \cdot\cdot & \ddots & \cdot\cdot \\
-\frac{v\pi}{L}\delta_{q,k}\delta_{q,0} + 2\Delta^{(1)}_{q,k}(t)\,u^{(a)}_{a,q}(0)u^{(a)}_{a,k}(0) & & \Delta^{(2)}_{q,k}(t)\,u^{(a)}_{a,q}(0)w^{(s)}_{s,k}(0) & \\
\cdot\cdot & \ddots & \cdot\cdot & \ddots \\
\hline
\ddots & \cdot\cdot & \ddots & \cdot\cdot \\
\Delta^{(2)}_{q,k}(t)\,u^{(a)}_{a,k}(0)w^{(s)}_{s,q}(0) & & -\frac{v\pi}{L}\delta_{q,k}\delta_{q,0} & \\
\cdot\cdot & \ddots & \cdot\cdot & \ddots
\end{array}
\right),
$$

$$
D = \left(
\begin{array}{c}
\vdots \\
\Gamma^{(1)}_{q}(t)u^{(a)}_{a,q}(0) \\
\vdots \\
\hline
\vdots \\
0 \\
\vdots
\end{array}
\right), \quad
E = \left(
\begin{array}{c}
\vdots \\
0 \\
\vdots \\
\hline
\vdots \\
\Gamma^{(2)}_{q}(t)w^{(s)}_{s,q}(0) \\
\vdots
\end{array}
\right).
\tag{85}
$$

The momentum indices $q, k$ run within the blocks demarcated by solid lines, whereas the sector indices $j = a, s$ change from one block to the other. The functions occurring above are defined via

$$
\Gamma^{(i)}_{q}(t) = \int_{0}^{L} dx\, g^{(i)}(x, t)\cos(qx),
\tag{86}
$$

$$
\Delta^{(i)}_{q,k}(t) = \int_{0}^{L} dx\, h^{(i)}(x, t)\cos(qx)\cos(kx) = \frac{1}{2}\left(h^{(i)}_{q+k}(t) + h^{(i)}_{|q-k|}(t)\right)
\tag{87}
$$

$$
h^{(i)}_{q}(t) = \int_{0}^{L} dx\, h^{(i)}(x, t)\cos(qx).
\tag{88}
$$

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
