# Peer review of "On the low-energy description for tunnel-coupled one-dimensional Bose gases"

_SciPost Physics, doi:SciPost Phys. 9, 025 (2020)_

## Round 1 · Referee Report · Anonymous · 2020-6-6

Strengths

1- It addresses a timely and relevant experimental question
2- Clearly written
3- It presents a clear and detailed introduction to the harmonic self-consistent method

Weaknesses

1-The theoretical setup has been proved to be insufficient in describing the experimental observations

Report

The authors provide an analytical study of an experimentally-relevant setup, namely two weakly coupled Bose-Einstein condensates through a strong barrier. This experiment is currently implemented and extensively studied in the Vienna’s group, greatly motivating any interesting result on this model.
At low energies, the system is known to be approximately described by the sine-Gordon field theory which the authors already studied in the self-consistent time–dependent harmonic approximation (SCTDHA) Ref. [67].
Here, the authors extend the previous setting including a more general dynamics in their self-consistent approximation, allowing for inhomogeneous field profiles and working in a hard-wall box potential. Compared with the previously studied periodic boundary conditions, boxes are expected to provide a better description of the experiment, which is realized in a harmonic trap. Most importantly, an additional correction to the Hamiltonian is considered, which couples the symmetric and antisymmetric sectors of the theory, allowing energy transfer between the two. In my understanding, this term has been included hoping it could capture the phase-locking observed in Ref. [24], however, as the authors write in the conclusions, this newly-introduced term seems to have mild effects on the dynamics and cannot provide a mechanism for the observed phase-locking.

In my opinion, the results discussed in this manuscript are correct, but from the technical point they only present an incremental advance compared with their previous publication Ref. [67]. Including the new interaction and inhomogeneities in the self-consistent harmonic approximation surely presents additional technical difficulties and calculations, but I could not appreciate the theoretical novelty of the method.
I think that the most interesting finding is that the coupling between the symmetric and antisymmetric sectors is not enough to describe the phase-locking, which I heard to be pointed out as the most probable culprit in several instances.
Even though the energy scale at which the SCTDHA can be applied is too small compared with that of the present experiment, I believe they are still close enough to trust the qualitative results and rule out the symmetric-antisymmetric coupling as the major character in the phase-locking mechanism.

In general, I think this manuscript presents results worth to be published in SciPost Physics, however I have two questions:

- The SCTDHA is supposed to hold at short time-scales where non-gaussian corrections grow, but I could not find in the paper an estimation of the validity time scales. Is there any quantity that can be computed within the SCTDHA in order to check its validity? It would be interesting to compare such a time scale with the phase-locking time of the experiment.

- I wonder if more realistic inhomogeneities can be captured building on the curved CFT introduced in SciPost Phys. 2, 002 (2017), including of course a tunneling term among the two condensates and resulting in an inhomogeneous sine-Gordon model. Can the authors comment on this?

Moreover, I would suggest the following improvements:

- In the introduction, the phase-locking mechanism is discussed and it appears to me as the main motivation of this work, however the effect of the new terms is discussed only in the conclusions. I think it could be helpful to mention straight from the beginning (i.e. in the introduction) the fact that the proposed extensions to the SCTDHA cannot reproduce the phase-locking mechanism.

- I think that the list of references on the FCS is incomplete. I know that the list of papers studying the FCS is very long, but I think at least the three references below must be included
Rev. Lett. 122, 120401 (2019).
SciPost Phys. 7, 072 (2019)
EPL 129 60007 (2020)

I also found some typos
incompatable->incompatible pg 3
poisson->Poisson pg 13
Ref. [63] is the same as Ref. [62]

I would recommend the authors to go through the manuscript once again and fix possible additional typos.

Requested changes

1- Anticipate in the introduction that the new terms cannot provide a mechanism for the observed phase-locking
2- Update the list of references
3- Fix minor typos in the manuscript

  • validity: high
  • significance: high
  • originality: good
  • clarity: high
  • formatting: excellent
  • grammar: excellent

Author:  Yuri Daniel van Nieuwkerk  on 2020-07-13  [id 883]

(in reply to Report 1 on 2020-06-06)

We thank the referee for their time in studying our work and for their useful comments. We have incorporated these comment in a new version of the manuscript in the following ways:

-We have added the suggested more recent references on full counting statistics .
-We have corrected the typos mentioned by the referee and done another check for any remaining errors.
-We have taken up the referee's advice to be more explicit about the fact that our suggested mixing term and hard-wall boundary conditions fail to capture the strong damping seen in experiment. To this end, we have added the following sentences to the penultimate paragraph of the Introduction:

"However, the observed effects are rather weak. This means that the presence of the box potential and the new sector-coupling term are insufficient to explain the experimentally observed damping phenomenon. At the same time, our results indicate that the simulation of a sine-Gordon model using the hard-wall box potential setup described here should not be severely restricted by the presence of the additional coupling term, which turns out to have only mild effects."

-We agree with the referee that it would be great to have an estimate of the timescales over which the SCTDHA is expected to hold. The SCTDHA neglects higher cumulants than the variance and is therefore a good approximation as long as these are small in the "full" theory. This also means that there is no way to estimate these higher cumulants within the SCTDHA. One would thus need to go beyond the SCTDHA to estimate its validity. This is numerically very costly for the full QFT and would constitute a separate research project. In our work Ref. [67], we therefore resorted to a comparison of the SCTDHA to numerically exact solutions for the zero mode, which plays a very important role in the Josephson oscillations. For that theory he SCTDHA was seen to give accurate results for the first ~3 density-phase oscillation periods, which is a long enough window to compare to experiments. We have therefore considered similar time scales in the current paper. To explain this, we have added the following sentences to the fourth paragraph of the Introduction:

"Our strategy is to treat the resulting perturbed sine-Gordon model in the self-consistent time-dependent harmonic approximation (SCTDHA) as described in \cite{Nieuwkerk2018b}. In that paper we have benchmarked the approximation using the dynamics of the zero modes in the antisymmetric sector only. It is the expectation value of these modes that displays the Josephson oscillations, making their dynamics vital to the problem at hand. Comparison to numerically exact results for this simplified problem indicated that the SCTDHA offers reliable results for early times corresponding to $\sim3$ density-phase oscillation periods. Based on those findings, we apply the approximation in the current work to the first few oscillation periods in the presence of sector coupling."

-We fully agree with the referee's remark that it would be interesting to include effects of a more realistic potential. In fact we did some work in this direction by following the approach of R Citro et al 2008 New J. Phys. 10 045011, which still allows one to work with a mode expansion in terms of known orthogonal polynomials. However, as explained by Brun and Dubail (SciPost Phys. 4, 37 (2018)) one needs to account for spatially dependent prefactors when considering the projections of physical observables to the low-energy subspace, which complicates matters. We therefore decided to focus on the simpler case of hard wall boundaries. We thank the referee for raising this point, and have added the following sentence to the last paragraph of the Conclusion:

"It would be interesting to combine such a microscopic approach with low-energy effective field theory calculations in the presence of such a quadratic trapping potential. This could be done by combining a SCTDHA with recent results \cite{Dubail2017,Brun2018} for inhomogeneous Luttinger liquids \cite{Gang03,OlDu03,Ghosh06,Citro08}."

---

## Round 7 · Referee Report · Anonymous (Referee 2) · 2020-7-24

Report

I am satisfied with the improved manuscript as well as with the answer to the questions addressed in my report, therefore I can recommend the paper for publication in Scipost Physics.

---

## Round 7 · List of Changes

-We have added the more recent references on full counting statistics suggested by the referee.
-We have corrected the typos mentioned by the referee.
-We have taken up the referee's advice to be more explicit in the Introduction about the fact that our suggested mixing term and hard-wall boundary conditions fail to capture the strong damping seen in experiment. To this end, we have added three sentences to the penultimate paragraph of the Introduction.
-We have added five sentences to the fourth paragraph of the Introduction to discuss the time scale over which the SCTDHA is expected to be accurate.
-We have added a sentence to the last paragraph of the Conclusion, mentioning the perspective to include the effects of a more realistic potential. We have added Refs. 79-84 to support this.

---

## Editorial Decision

published